# Modeling succinate dehydrogenase loss disorders in *C. elegans* through effects on hypoxia-inducible factor

Megan M. Braun[¤a], Tamara Damjanac[¤b], Yuxia Zhang[¤c], Chuan Chen, Jinghua Hu, L. James Maher III[ID] *

Department of Biochemistry and Molecular Biology, Mayo Clinic College of Medicine and Science, Rochester, MN, United States of America

¤a Current address: University of Wisconsin–Madison Neuroscience Training Program, Madison, WI, United States of America
¤b Current address: University of Minnesota Medical School–Twin Cities, Minneapolis, MN, United States of America
¤c Current address: MD Anderson Cancer Center, the University of Texas, Dallas, TX, United States of America
* maher@mayo.edu

**Data Availability Statement:** All relevant data are within the manuscript.

## Abstract

Mitochondrial disorders arise from defects in nuclear genes encoding enzymes of oxidative metabolism. Mutations of metabolic enzymes in somatic tissues can cause cancers due to oncometabolite accumulation. Paraganglioma and pheochromocytoma are examples, whose etiology and therapy are complicated by the absence of representative cell lines or animal models. These tumors can be driven by loss of the tricarboxylic acid cycle enzyme succinate dehydrogenase. We exploit the relationship between succinate accumulation, hypoxic signaling, egg-laying behavior, and morphology in *C. elegans* to create genetic and pharmacological models of succinate dehydrogenase loss disorders. With optimization, these models may enable future high-throughput screening efforts.

## Introduction

Mitochondrial disorders can result from mutations affecting enzymes of oxidative metabolism [1–4]. Interestingly and surprisingly, some cancers are caused by gain-of-function or loss-of-function mutations of genes encoding metabolic enzymes in susceptible tissues [5]. For example, paraganglioma and pheochromocytoma (PPGL) are rare neuroendocrine tumors [6–8] that originate in the parasympathetic and sympathetic ganglia, are highly angiogenic, and may secrete catecholamines. Up to 30% of PPGL tumors are hereditary [9].

All four subunits of the mitochondrial enzyme succinate dehydrogenase (SDH) have been identified as tumor suppressors in familial PPGL[10–13], with loss of heterozygosity accounting for tumorigenesis. The succinate accumulation hypothesis attributes tumorigenesis following SDH loss to an oncometabolite role for excess succinate [14]. Loss-of-function mutations of SDH subunits lead to dysfunctional complexes [15, 16]. The resulting TCA cycle dysfunction drives metabolic remodeling with dependence on glycolysis [17] and a profound

**Funding:** This work was supported by the Mayo Foundation, the Paradifference Foundation, and by NIH grant CA166025 (LJM). Some C. elegans strains were provided by the CGC, which is funded by the NIH Office of Research Infrastructure Programs (P40 OD010440). The funders had no role in study design, data collection and analysis, decision to publish, or preparation of the manuscript.

**Competing interests:** The authors have declared that no competing interests exist.

accumulation of succinate as defective SDH cannot oxidize this dicarboxylic acid to fumarate. Excess succinate acts as a competitive inhibitor of enzymes belonging to the 2-ketoglutarate-dependent dioxygenase family. This family of iron-dependent enzymes, numbering more than 40 in humans [18], catalyzes oxidation reactions splitting molecular oxygen to incorporate one oxygen atom into the substrate with oxidative decarboxylation of co-substrate, 2-ketoglutarate, to form succinate [19].

Since many enzymes belong to the 2-ketoglutarate-dependent dioxygenase family, there are many potential consequences of succinate accumulation upon SDH loss [20]. One susceptible enzyme of interest is HIF-α prolyl-hydroxylase (PHD), which participates in the oxygen sensing mechanism of animals. Under normoxic conditions, PHD hydroxylates HIF-α transcription factor subunits, marking the proteins for polyubiquitination by von Hippel-Lindau protein and eventual degradation by the proteasome [21]. Under hypoxic conditions, molecular oxygen is limiting so the PHD-catalyzed dioxygenase reaction slows and HIF-α subunits avoid degradation and translocate to the nucleus to interact with constitutively-expressed HIF-β. The resulting transcription factors activate genes driving angiogenesis and glycolysis to adapt to hypoxia. High levels of succinate inhibit PHD, creating a pseudohypoxic condition that is hypothesized to be tumorigenic in susceptible cell types [14]. It remains unknown how succinate poisoning of PHD and/or other 2-ketoglutarate-dependent dioxygenases drives tumorigenesis. In the absence of rodent models and SDH-loss PPGL cell lines, understanding the linkage between SDH loss and tumorigenesis is an urgent and unmet need.

Our limited understanding of the mechanistic impact of SDH loss on cellular processes and tumorigenesis has thwarted PPGL therapeutic advances. We therefore sought to establish a *C. elegans* model embodying genetic and biochemical aspects of SDH-loss disorders including PPGL. The soil nematode *C. elegans* provides an inexpensive, easily-maintained, genetically-tractable model organism with a fully-sequenced genome [22, 23]. Moreover, fully 40% of genes known to be associated with human diseases have clear *C. elegans* orthologs. For example, while humans have three HIFα subunits (HIF-1α, HIF-2 α, and HIF-3α) encoded by three separate genes, *C. elegans* has only a single hif-1 α gene, facilitating conclusive genetic analysis [24]. In principle, changes in *C. elegans* phenotype or behavior associated with mutations related to SDH and HIF function could create models for high-throughput screening of compounds that suppress or exacerbate these characteristics in intact animals. Whole-animal suppression screens have the advantage of simultaneously monitoring efficacy and toxicity.

Inspiration for a *C. elegans* model of the molecular changes associated with SDH-loss disorders such as PPGL came from the previous fascinating observation that mutation of *egl-9 (sa307)*, the *C. elegans* ortholog of human PHD, unexpectedly causes increased egg retention in hermaphrodite worms [25]. In retrospect, it seems likely that oxygen-sensing in egg-laying behavior is adaptive, suppressing egg-laying in inhospitable environments. The egg-laying defect is HIF-1-dependent, as *egl-9(sa307):hif-1(ia4)* double mutants and *hif-1(ia4)* single mutants both exhibit normal egg laying behavior [26].

Here we exploit HIF-1-dependent egg retention to create *C. elegans* models of SDH-loss human PPGL. We hypothesized that cellular changes impinging on the *C. elegans* HIF pathway should be revealed in the egg retention phenotype because, as noted, succinate accumulation upon SDH loss inhibits PHD activity. To investigate this, we utilized both genetic (cell-specific gene expression or knockdown) and pharmacological [treatment with dimethyloxalylglycine (DMOG), a cell-permeable succinate analog] approaches. Further, we report a novel image-based readout of worm morphology with the potential to monitor egg retention phenotypes more efficiently in drug screening. With further optimization, this work suggests a path toward future screening for non-toxic small molecules that suppress the egg retention phenotype caused by succinate inhibition of EGL-9. Such agents might function to therapeutically relieve

succinate inhibition of 2-ketoglutarate-dependent dioxygenases in PPGL and other SDH-loss disorders.

## Materials and methods

### Strains and maintenance

Previously described *C. elegans* strains utilized in this study include N2, JT307 *egl-9 (sa307)*, CB6088 *egl-9(sa307):hif-1(ia4)*, and ZG31 *hif-1(ia4)*. These strains were obtained from the *Caenorhabditis* Genetics Center.

*C. elegans* were grown and maintained on nematode growth media (NGM) agar seeded with *E. coli* OP50 at room temperature unless otherwise noted. Standard alkaline sodium hypochlorite treatment was used to establish synchronous populations of worms for egg counting and imaging studies. For egg counting studies, individual worms were placed in bleach droplets and eggs were counted after cuticle dissolution [27]. Media components were obtained from Sigma.

### Generation of transgenic lines

The *C. elegans Punc-31* promoter was used to drive expression of Hif transgenes or RNA interference constructs for SDH subunit knockdown, allowing these effects to be limited to neurons known to be important for egg-laying. DNA oligonucleotides were obtained from IDT. *Punc-31* was amplified by PCR from *C. elegans* genomic DNA (forward primer sequence 5′ –AACA ACTTGGAAATGAAATACGAGAACTTAAACCATTAAA; reverse primer sequence 5′ –GACCT GCAGGCATGCAAGCTGATGTTCCAAACGAAGACTG) and Gibson assembly was used to insert *Punc-31* into HindIII-linearized pPD95_75. Following ligation and cloning, the resulting plasmid was linearized by BamHI cleavage 30 bp downstream from the *Punc-31* insertion and *egl-9(+)* coding sequence that had been amplified by PCR from genomic DNA (forward primer sequence 5′ – CCTGCAGGTCGACTCTAGAGCACATGACATGAGCAGTGCCCCAAATGA; reverse primer sequence 5′ –CTTTGGCCAATCCCGGGGATCGATGTAATACTCTGGGTTTG) or *hif-1(+)* coding sequence that had been amplified by PCR from genomic DNA (forward primer sequence 5′ –CCTGCAGGTCGACTCTAGAGATCAAGATGGAAGACAATCG; reverse primer sequence 5′ – CTTTGGCCAATCCCGGGGATCAGAGAGCATTGGAAATGGGG) was inserted using a second round of Gibson assembly. Constructs jhuEx[*Punc-31*::*EGL-9*] and jhuEx[*Punc-31*::*HIF-1*] were co-injected into adult hermaphrodite *egl-9(sa307)* and *egl-9 (sa307):hif-1(ia4)* worms, respectively, along with plasmid *pRF4* encoding a mutant collagen (*rol-6*(su1006)) that induces a dominant "roller" phenotype [28, 29]. Importantly, in preliminary experiments it was demonstrated that the *rol-6* marker does not affect egg-laying behavior or egg retention.

*Punc-31* was amplified by PCR with using a different set pair of primers (forward primer sequence 5′ – ATGACCATGATTACGCCACGAGAACTTAAACCATTAAATA; reverse primer sequence 5′ –CCTGCAGGCATGCAAGCTGATGTTCCAAACGAAGACTGCA) for Gibson assembly into HindIII-linearized pPD49_78. Sense and antisense domains of a 496-bp region of the *C. elegans sdhb-1* gene were assembled from appropriate primers by PCR from genomic DNA targeting parts of exons 1 and 3 (44 and 121 bp respectively) and all of exon 2 (sense forward primer sequence 5′ – AGCTTGCATGCCTGCAATCGTTTCAACCCAGAAGCACCAG; sense reverse primer sequence 5′ – CAAAGTGTGGCTGAACGTGACACGTTCAGCCACACTTTGG; antisense forward primer sequence 5′ – CCAAGTGTGGCTGAACGTGTCACGTTCAGCCACA CTTTG; antisense reverse primer sequence 5′ – GATCCTCTAGAGTCGACCTCGTTTCAACC CAGAAGCACCA). Sbf1 was used to linearize the resulting plasmid 16 bp downstream from the *Punc-31* insertion. A second round of Gibson assembly was used to insert the sense and

antisense *sdhb-1* segments into the SbfI-linearized plasmid, forming an inverted repeat encoding a long RNA hairpin for RNA interference. The resulting plasmid jhuEx[*Punc-31::sdhb-1* (IR)] was co-injected with pBX into adult hermaphrodite N2 worms.

## DMOG treatment

After synchronization, worm concentration was approximated by counting the number of worms in ten 10-µL drops of medium. Culture volume was diluted to 100 worms/mL. One mL of culture was then added to each well of a 12-well plate. OP50 bacteria were added at 5 mg/mL. Plates were sealed with an aluminum plate sealer and transferred to a room temperature shaker.

Worms received the first treatment with dimethyloxalylglycine (DMOG; Sigma), a water-soluble succinate analog, approximately four hours after culture initiation. On the following days, DMOG was added and replenished twice daily at 8-hour intervals to account for spontaneous hydrolysis. It was assumed that each dose of compound was hydrolyzed during each interval, so DMOG treatment concentration is termed "nominal."

## Acquisition and analysis of *C. elegans* images

*C. elegans* worms were transferred from liquid culture onto clean NGM plates and rinsed with s-complete medium. Adults were manually separated from larvae with a worm pick onto fresh NGM plates. Digital brightfield images were obtained manually using a Leica DMi1 camera using the 10x objective and converted to grayscale tiff files using Adobe Photoshop. WormSizer [30], an open source plugin compatible with Fiji [31], was used to obtain length and width measurements for each imaged animal.

## Statistical analysis

Values are expressed as mean +/- standard deviation for the indicated number of independent experiments. The statistical analysis was performed using the Student's t-test or a one way analysis of variance (ANOVA) test with post-hoc Tukey HSD, or a Dunnett's test with R Studio software. A P value of less than 0.05 was considered statistically significant.

## Results

### Cell-specific knockdown of SDHB-1 leads to increased egg retention in N2 worms

We set out to determine whether *C. elegans* can be used as a genetic model of the SDH-loss cells present in human mitochondrial disorders including familial PPGL tumors. RNAi screens have shown that systemic knockdown of any of the four SDH subunits (SDHx) is embryonic lethal in *C. elegans* [32, 33], as in mammals. Seeking screenable phenotypes associated with SDH loss in worms, it was therefore necessary to limit SDHx knockdown to a subpopulation of cells consistent with viability. We hypothesized that egg-laying behavior controlled by EGL-9 would be sensitive to succinate accumulation such that succinate inhibition of EGL-9 would phenocopy EGL-9 loss and drive egg retention. The *unc-31* promoter (*Punc-31*) was selected to drive expression of test genes because this promoter has been shown to be active in neurons, including those believed to be responsible for egg laying [34]. To test this, we constructed two transgenic lines based on known egg-laying behavior. Hermaphrodite worms homozygous for the *egl-9(sa307)* mutation retain eggs. We found that *Punc-31*-driven expression of functional EGL-9 in *egl-9(sa307)* mutants significantly relieved egg retention from *egl-9(sa307)* mutant levels (P<2.62e-14) to near wild type levels (mean eggs per worm

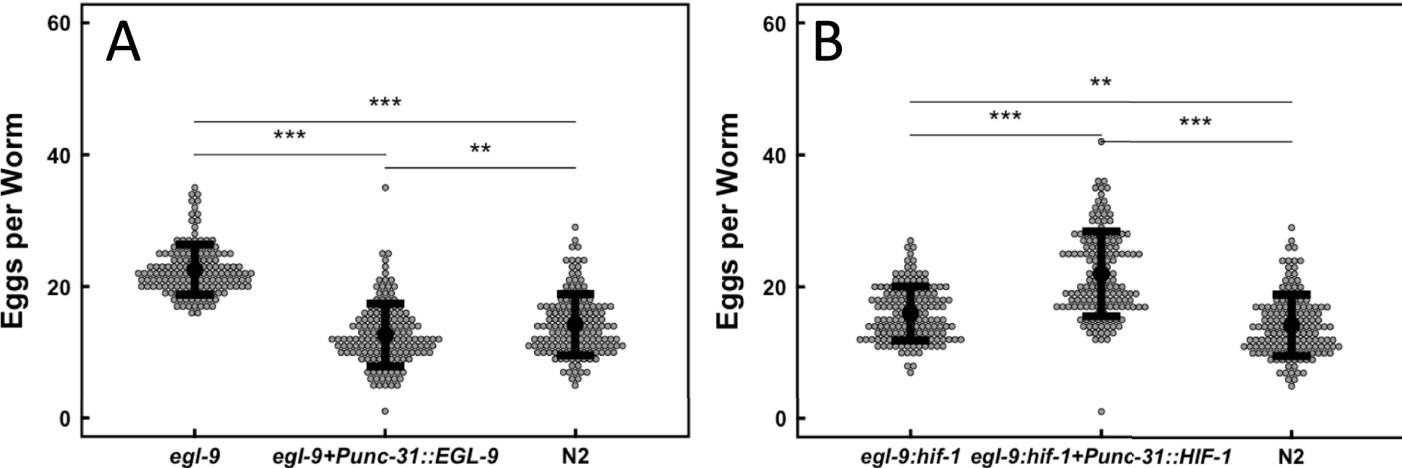

**Fig 1. *Punc-31* control of *egl-9* and *hif-1* expression is adequate to control egg retention phenotype.** A. *Punc-31*-driven expression of EGL-9 in *egl-9(sa307)* mutants is sufficient for statistically significant reduction of egg retention below egl-9(sa307) levels (P<2.62e-14) to near wild type levels (mean eggs per worm are 14.2 and 12.7 respectively; P = 0.007). B. *Punc-31*-driven expression of HIF-1 in *egl-9(sa307):hif-1(ia4)* double mutants significantly increases egg retention above *egl-9(sa307):hif-1 (ia4)* and N2 levels (P<2.62e-14 in both cases), to near *egl-9(sa307)* egg retention levels. ANOVA with post-hoc Tukey's HSD test was used to compare mean egg retention. **P<0.01, ***P<0.001. Error bars show standard deviation.

are 14.2 and 12.7 respectively; P = 0.007; Fig 1A). Likewise, *Punc-31*-driven expression of wild type *hif-1(+)* in *egl-9(sa307):hif-1(ia4)* hermaphrodites increased egg retention significantly above wild type and *egl-9(sa307):hif-1(ia4)* double mutants levels (P<2.62e-14 for both cases; Fig 1B). These results demonstrate that the *unc-31* promoter defines a cell compartment that controls egg-laying behavior. We considered if the more limited neuroendocrine cell compartment defined by *tdc-1* promoter (*Ptdc-1*) activity might also be adequate to control egg-laying behavior. *Ptdc-1* activity is thought to be limited primarily to the four uv1 neuroendocrine cells of *C. elegans*, known to play a prominent role in hormonal control of egg laying. Interestingly, in contrast to *Punc-31*, we found that *Ptdc-1*-driven expression of functional *hif-1(+)* in *egl-9(sa307):hif-1(ia4)* mutants was inadequate to induce egg retention (data not shown). It is unknown whether this result is due to the tissue restriction of *Ptdc-1* or its strength.

Based on these results, *Punc-31* was chosen to drive cell-specific knockdown of SDHB-1 by RNA interference after injection of a plasmid containing an *sdhb-1* inverted repeat (IR) under the control of *Punc-31*. Disruption of the SDH complex in unc-31-expressing cells is hypothesized to mimic essential biochemical phenotypes of SDH-loss disorders such as PPGL. Three stable worm lines were generated. After synchronization, all individuals carrying the *sdhb-1* RNAi transgene showed increased egg retention relative to controls (Fig 2). This result demonstrates for the first time that SDH function in *Punc-31*-positive cells is necessary for normal egg-laying behavior. We hypothesize that SDH knockdown results in intracellular succinate accumulation, known to inhibit 2-ketoglutarate-dependent dioxygenases such as EGL-9. According to this model, EGL-9 inhibition prevents HIF-1 hydroxylation, stabilizing HIF-1 and promoting HIF-1 signaling and egg retention behavior in *C. elegans*. We note that global succinate accumulation in whole worms is not expected for SDH knockdown under these conditions, as effects would be limited to the small subset of cells where *Punc-31* is active.

### DMOG treatment increases egg retention in N2 worms

To test the hypothesis that succinate accumulation alone is sufficient to drive egg retention in *C. elegans*, we studied egg-laying behavior in the presence of dimethyloxalylglycine (DMOG),

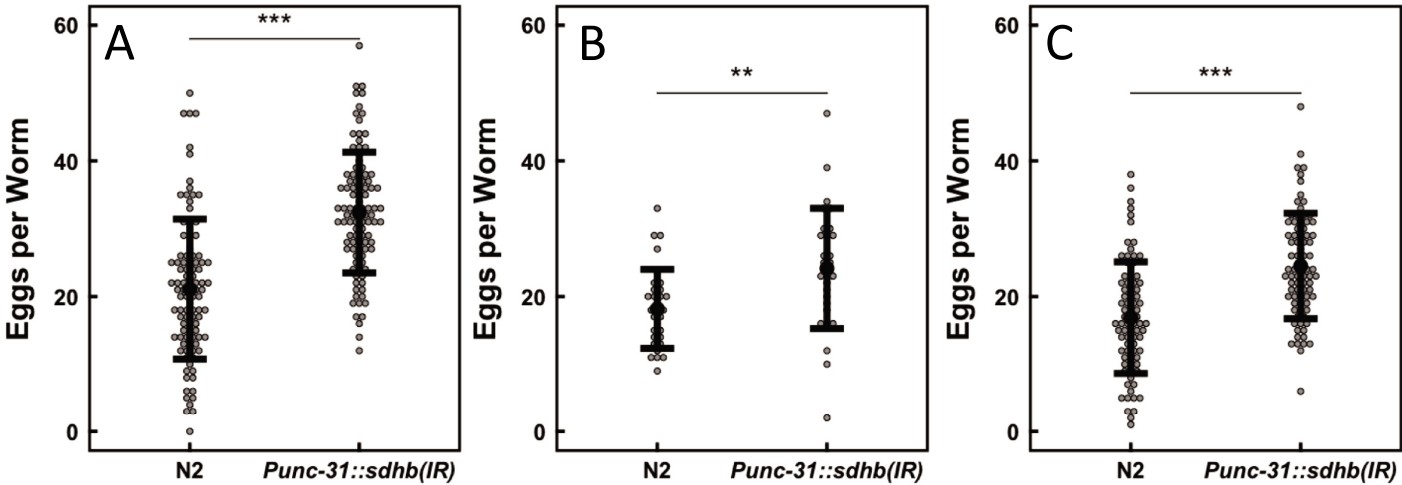

**Fig 2. Knockdown of SDHB-1 in *unc-31*-expressing cells increases egg retention in three independently derived worm lines.** *sdhb-1* knockdown was accomplished by microinjection of a plasmid containing *Punc-31* driven expression of a 496-bp *sdhb-1* fragment cloned directly before an inverted repeat of the sequence. An independent Student's T test was used to compare means. **P<0.01, ***P<0.001. Error bars show standard deviation.

a cell-permeable succinate analog. DMOG is the prodrug of N-oxalylglycine (NOG), which is known to inhibit 2-ketoglutarate-dependent dioxygenases but is unable to permeate cell membranes [35]. Previous studies have shown that mammalian cells treated with DMOG show an increase in transcription of HIF-1-responsive genes [36]. Consistent with our observations for *sdhb-1* knockdown, treatment of *C. elegans* hermaphrodites with DMOG induced egg retention in wild type N2 worms, but not in *hif-1(ia4)* worms (Fig 3). The implications of these results are two-fold. First, DMOG-induced egg retention in N2 worms provides a second

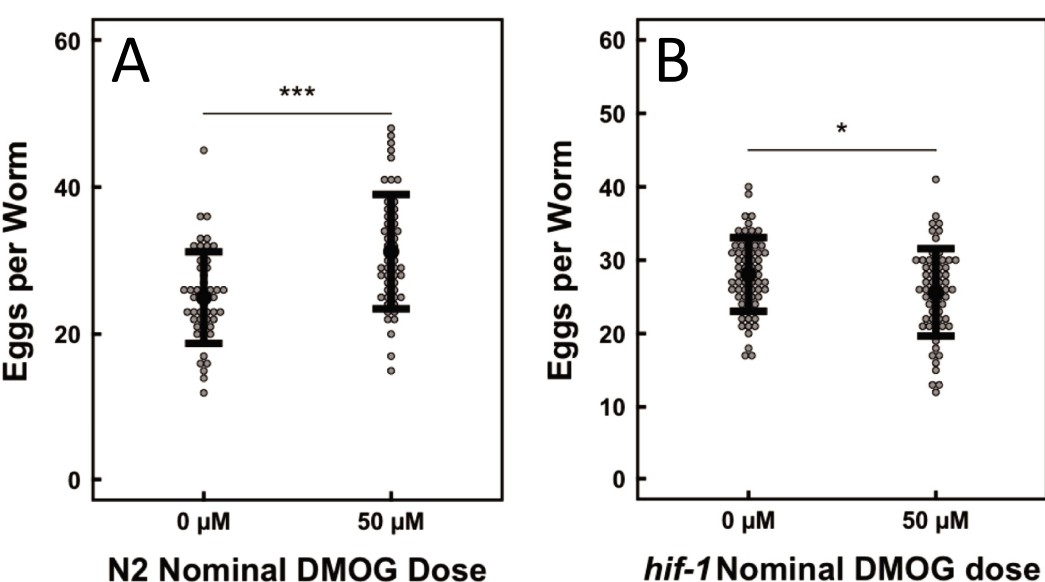

**Fig 3. DMOG treatment impacts egg retention in a strain-specific manner.** A. DMOG treatment increases egg retention in N2 worms. B. No egg retention defect was observed in DMOG-treated *hif-1(ia4)* worms. An independent Student's T test was used to compare means. *P<0.05, **P<0.01, ***P<0.001. Error bars show standard deviation.

SDH-loss PPGL model based on a cell-permeable metabolite analog. Second, the inability of DMOG to affect egg laying in *hif-1(ia4)* mutants demonstrates the HIF-1-dependence of this chemical mechanism of egg retention in N2 worms. This observation supports a model attributing egg retention in N2 worms to increased HIF-1 signaling resulting from DMOG inhibition of EGL-9.

## DMOG treatment alters N2 worm body morphology

*C. elegans* egg retention studies are commonly performed manually either by observing the eggs in an intact animal or after dissolving the cuticle in sodium hypochlorite and counting the eggs in resistant clutches [27]. To more quickly gather egg retention data in a manner that might be optimized in the future for possible high-throughput screening of agents that alter this phenotype, we sought a quantitative surrogate for egg retention. Assays of chitinase release have previously been reported for this purpose[37], but were found to be too variable and imprecise for our purpose. Changes in body morphology were then considered because egg retention might reasonably be expected to affect girth. The WormSizer software tool was used to collect a variety of measurements from brightfield images comparing synchronized untreated and DMOG-treated N2 worms [30]. As hypothesized, DMOG treatment was observed to increase the girth of worms, and this effect was dose-dependent (Fig 4A). Intriguingly, DMOG treatment also decreased length of N2 worms in a dose-dependent manner due to unknown mechanisms (Fig 4B). Thus, DMOG-treated worms were both wider and shorter than normal as evidenced by a reproducible dose-dependent decrease in width:length ratio (Fig 4C). This observation suggests that optimization could lead to a new image-based screening approach for phenotypes related in egg retention in *C. elegans*.

## Discussion

There is increasing interest in understanding disorders caused by cellular metabolite imbalances [18]. Of particular importance to us are cancers driven by alteration of metabolic enzymes, resulting in accumulation of dicarboxylates such as succinate, fumarate, and 2-hydroxyglutarate. These oncometabolites inhibit 2-ketoglutarate-dependent enzymes important for many aspects of cell regulation, including hypoxic response, and epigenetic

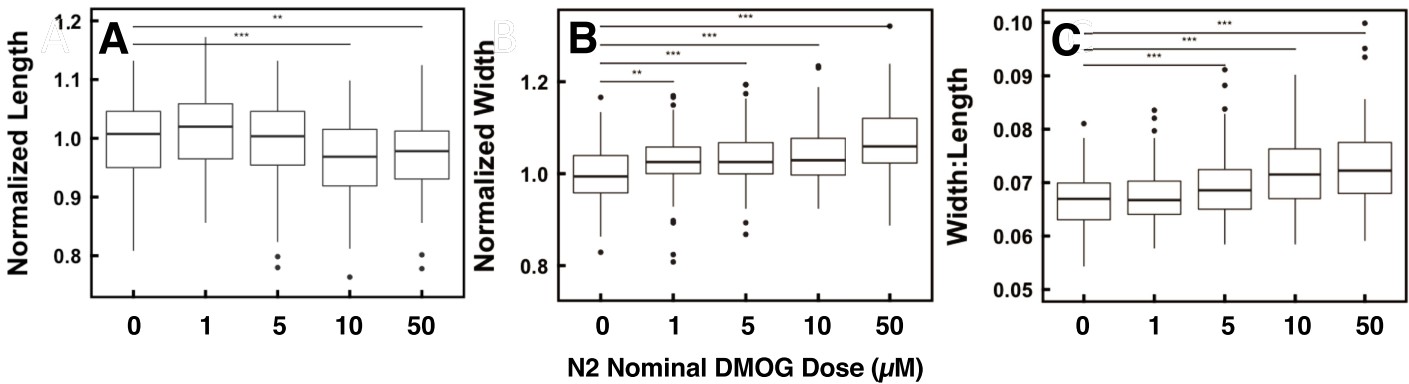

**Fig 4. DMOG treatment affects worm body morphology in a dose-dependent manner.** A. Midpoint width increases with increasing DMOG dose. B. Length decreases with increasing DMOG dose. C. Width:length ratio increases with increasing DMOG dose. Worms treated with higher DMOG doses are increasingly shorter and fatter than untreated worms. Brightfield images were analyzed as described in Methods. Measurements were normalized to untreated worm measurements. A Dunnett's test was used to compare mean morphological characteristics of DMOG-treated to untreated worms. **$P<0.01$, ***$P<0.0001$. Boxplot indicates the median, first and third quartiles and whiskers indicate the largest value within $1.5 \times$ interquartile range. Data points beyond $1.5 \times$ interquartile range are plotted individually.

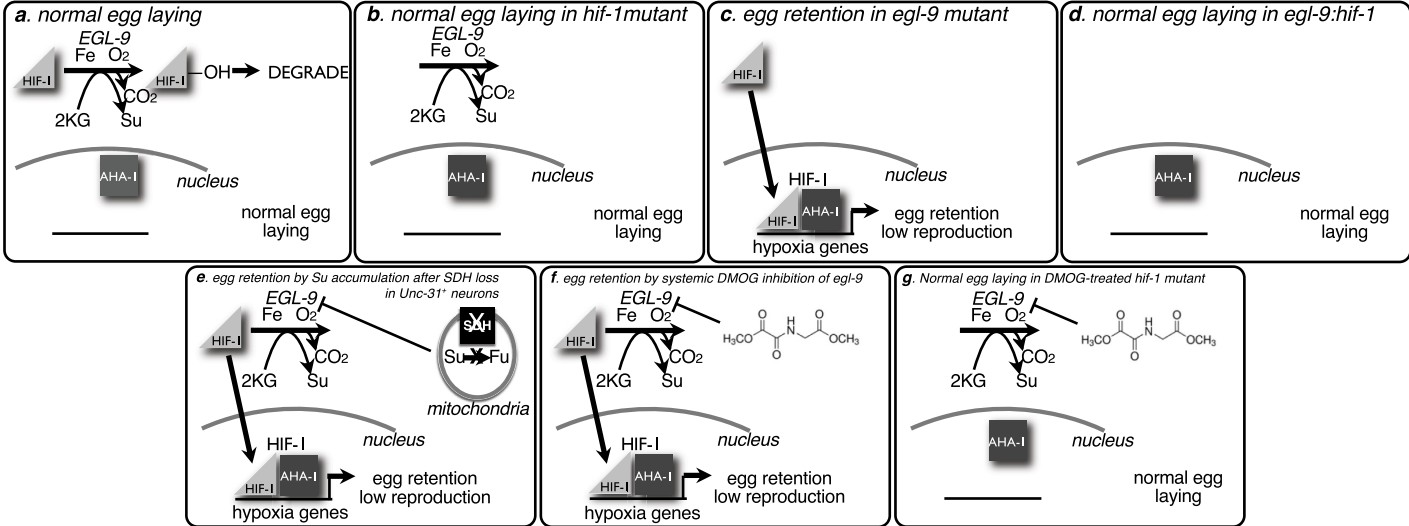

**Fig 5. Summary of EGL-9 activity and impact on egg laying in *C. elegans* models described here.** A. In normoxia, EGL-9 prolyl-hydroxylase marks HIF-1 for degradation and egg-laying is normal. B. *hif-1* mutant worms lack stable HIF-1 and display normal egg laying. C. *egl-9* mutant worms cannot hydroxylate HIF-1. Stable HIF-1 translocates to the nucleus, and interacts with AHA-1, serving as a transcription factor for hypoxia genes. These worms show increased egg retention. D. *hif-1* mutation rescues the egg-laying phenotype in *egl-9:hif-1* double mutant worms: though EGL-9 activity is missing, the corresponding absence of HIF-1 precludes HIF-1 signaling and egg laying is normal. E. *Punc-31* driven knockdown of SDHB-1 leads to a succinate accumulation that inhibits EGL-9 activity thus stabilizing HIF-1 and leading to egg retention. F. DMOG inhibits EGL-9 activity, stabilizing HIF-1 and leading to egg retention in N2 worms. G. DMOG can also inhibit EGL-9 activity in *hif-1* worms but no HIF-1 signaling occurs leading to normal egg laying.

regulation through demethylation of histones, DNA, and RNA [38, 39]. Studies to develop potential therapies for SDH-loss disorders, including familial PPGL, have been limited by the absence of cell and animal models of the SDH-loss condition [40]. We have previously exploited SDH-loss yeast models for drug screening to identify vulnerabilities induced by SDH loss and succinate accumulation [41]. Here we envision a different approach–the potential for a suppression screen in intact *C. elegans* worms where a measurable quantitate phenotype reflects oncometabolite accumulation. Such a system would allow screening of drug libraries for non-toxic agents that suppress the phenotype driven by SDH loss and succinate accumulation. Such agents might function by preventing or discharging succinate accumulation.

Toward this end we report both genetic and chemical *C. elegans* models that link quantifiable egg-laying phenotypes to SDH loss and succinate accumulation. These models include egg retention secondary to SDH loss in *Punc31*[+] cells, and egg retention secondary to whole-body treatment with succinate analog DMOG. We further show that worm body morphology changes in a dose-dependent manner with DMOG treatment, paralleling egg retention, and providing a possible future approach for high-content image screening of worm phenotypes if the methodologies can be optimized. These results and their interpretations are summarized in Fig 5.

The results described here open the possibility that *C. elegans* can be applied after future assay optimization to high-throughput screening of chemical libraries for non-toxic agents that suppress effects of SDH loss and succinate accumulation. Such agents would reveal druggable pathways that might be altered to block oncometabolite effects in cancers of interest.

## Acknowledgments

We acknowledge the technical assistance of Nicole Becker, Artie Sletten, Alex Blee, and Deb Evans.

## Author Contributions

**Conceptualization:** Megan M. Braun, Jinghua Hu, L. James Maher III.

**Data curation:** Megan M. Braun, Tamara Damjanac.

**Formal analysis:** Megan M. Braun, L. James Maher III.

**Funding acquisition:** Jinghua Hu, L. James Maher III.

**Investigation:** Megan M. Braun, Tamara Damjanac.

**Methodology:** Megan M. Braun, Tamara Damjanac, Yuxia Zhang, Chuan Chen.

**Project administration:** L. James Maher III.

**Supervision:** Yuxia Zhang, Chuan Chen, Jinghua Hu, L. James Maher III.

**Validation:** Megan M. Braun, Tamara Damjanac, Jinghua Hu.

**Visualization:** Jinghua Hu, L. James Maher III.

**Writing – original draft:** Megan M. Braun, L. James Maher III.

**Writing – review & editing:** Megan M. Braun, Jinghua Hu, L. James Maher III.

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
