## [Decision Letter · Decision Letter 0]

25 Nov 2019

PONE-D-19-29380

Modeling succinate dehydrogenase loss disorders in C. elegans

PLOS ONE

Dear Dr. Maher, III,

Thank you for submitting your manuscript to PLOS ONE. After careful consideration, we feel that it has merit but does not fully meet PLOS ONE’s publication criteria as it currently stands. Therefore, we invite you to submit a revised version of the manuscript that addresses the points raised during the review process.

We would appreciate receiving your revised manuscript by Jan 09 2020 11:59PM. To enhance the reproducibility of your results, we recommend that if applicable you deposit your laboratory protocols in protocols.io, where a protocol can be assigned its own identifier (DOI) such that it can be cited independently in the future. For instructions see: http://journals.plos.org/plosone/s/submission-guidelines#loc-laboratory-protocols

We look forward to receiving your revised manuscript.

Kind regards,

Ales Vicha, M.D., PhD

Academic Editor

PLOS ONE

Journal Requirements:

1. To comply with PLOS ONE submission guidelines, in your Methods section, please provide additional information regarding your statistical analyses. For more information on PLOS ONE's expectations for statistical reporting, please see https://journals.plos.org/plosone/s/submission-guidelines.#loc-statistical-reporting.

2. At this time, we ask that you please provide the source of the primers and NGM media used in this study.

3. Thank you for including the following funding information within your acknowledgements section; "This work was supported by the Mayo Foundation, the Paradifference Foundation, and by NIH grant CA166025 (LJM). Some C. elegans strains were provided by the CGC, which is funded by the NIH Office of Research Infrastructure Programs (P40 OD010440). "

Additional Editor Comments (if provided):

The present article focuses on exploring the relationship between succinate accumulation, hypoxic signaling, egg-laying behavior, and morphology in C. elegans. In the present study, authors specifically focus on egg retention in C. elegans associated with SDH-loss disorders. They hypothesize that HIF-1alpha is the main player in egg retention. Since almost entire manuscript is devoted to this problem, the title of the manuscript should be more specific. This article is of interest to the readers of this journal.

The reviewer has a few comments:

1. Why was only the HIF-1alpha but not HIF-2alpha was proposed in the hypothesis? It is known that SDH loss is associated with HIF-2alpha changes.

2. What is the mechanism how Punc-31 affects SDHB expression? The page number 6 is very hard to read and understand. The same to applies to Figure 5.

3. Succinate measurement is not included, what is the reason for this?

4. HIF downstream genes should be shown in a way to see the effect of HIF-1alpha and HIF-2alpha…

5. How is this work related to paraganglioma? While SDH is, this work may not be and therefore some information related to paraganglioma and therapies/incidence could be deleted. This information can be also reduced in the summary.

6. References 6,7 and 9 should include original studies.

7. von Hippel Lindau should be von Hippel-Lindau

Reviewers' comments:

Reviewer's Responses to Questions

**Comments to the Author**

1. Is the manuscript technically sound, and do the data support the conclusions?

Reviewer #1: Yes

Reviewer #2: Yes

2. Has the statistical analysis been performed appropriately and rigorously? 

Reviewer #1: Yes

Reviewer #2: Yes

3. Have the authors made all data underlying the findings in their manuscript fully available?

Reviewer #1: Yes

Reviewer #2: Yes

4. Is the manuscript presented in an intelligible fashion and written in standard English?

Reviewer #1: Yes

Reviewer #2: Yes

5. Review Comments to the Author

Reviewer #1: It is clear form the paper that:

- SDHB KD is associated with egg retention

- This finding is linked to succinate through regulation of HIF1 stabilization

- Evaluation of girth provides an indirect reliable marker of succinate accumulation

In this way the C. Elegans model could be a useful tool for screening drugs

1- A critical issue of this paper for translating results into PPGL would be that the link between PPGL tumorigenesis and HIF1 is still debated. Many experiments suggest that HIF2 is probably more important than HIF1. Is HIF2 also involved in egg laying behavior ? Please comment.

2- Although intracellular succinate is very important in SDHB-related PPGL, it seems to be not only contained into the mitochondria and cytosol of the mutated cells but could act as an extracellular mediator, possibly via a specific receptor '"hormone like action". Did the authors assess the concentrations of intra-cellular and extra-cellular succinate (media) in SDHB KD models ? Please add informations. Would the results be similar if the media is removed very often during experiments ? Please add informations.

3- Succinate accumulation may lead to acidific pH. Is pH controlled during experiments ? Is extra or intracellular pH influences egg laying behavior ? Please add informations.

Reviewer #2: This is an interesting manuscript documenting a link between SDH and biological activities of C. elegans. The link between one of the key enzymes of the Krebs cycle, SDH, that feeds electron into the OXPHOS system, and biology of a worm is intriguing.

Concerning the experimental set-up, the manuscript is well executed, although I have some concerns about presentation of results. Figure 5 is really difficult to read. Although there are significant differences, the individual data are too close to each other, so the authors may start the y-scale at the value of 0.5 or 0.75 rather than 0.

The Introduction needs to be re-written partially. It is a bit too long. Also, the authors should refer to a key recent review that documents the role of SDH (complexII, CII) in the Krebs cycle and the OXPHOS system, i.e. Bezawork-Geleta A et al. Mitochondrial complex II: At the crossroads. Trends Biochem Sci 2017, 42, 312-325.

The authors state in the Introduction that 'Loss-of-function mutation of any SDH subunit leads to a nonfunctional complex...' This is not exactly true. For example. loss of SDHB results in the assembly of the so called CII-low, a subcomplex of CII comprising SDHA and SDHAF2 and/orSDHAF4 subunits. This CII-low allows for slow rate tumour formation> This should be included in the manuscript, as well as the original reference, where this was shown, i.e. Bezawork-Geleta A et al. Alternative assembly of respiratory complex II connects energy stress to metabolic checkpoints. Nat Commun 2018, 9, 2221.

6. PLOS authors have the option to publish the peer review history of their article (what does this mean?). If published, this will include your full peer review and any attached files.

Reviewer #1: No

Reviewer #2: No

---

## [Author Response · Author response to Decision Letter 0]

3 Dec 2019

See cover letter and response to reviewers

---

## [Decision Letter · Decision Letter 1]

12 Dec 2019

Modeling succinate dehydrogenase loss

disorders in C. elegans through effects on

hypoxia-inducible factor

PONE-D-19-29380R1

Dear Dr. Maher, III,

We are pleased to inform you that your manuscript has been judged scientifically suitable for publication and will be formally accepted for publication once it complies with all outstanding technical requirements.

With kind regards,

Ales Vicha, M.D., PhD

Academic Editor

PLOS ONE

Additional Editor Comments (optional):

Reviewers' comments:

Reviewer's Responses to Questions

**Comments to the Author**

1. If the authors have adequately addressed your comments raised in a previous round of review and you feel that this manuscript is now acceptable for publication, you may indicate that here to bypass the “Comments to the Author” section, enter your conflict of interest statement in the “Confidential to Editor” section, and submit your "Accept" recommendation.

Reviewer #1: All comments have been addressed

Reviewer #2: All comments have been addressed

2. Is the manuscript technically sound, and do the data support the conclusions?

Reviewer #1: Yes

Reviewer #2: Yes

3. Has the statistical analysis been performed appropriately and rigorously? 

Reviewer #1: Yes

Reviewer #2: Yes

4. Have the authors made all data underlying the findings in their manuscript fully available?

Reviewer #1: Yes

Reviewer #2: Yes

5. Is the manuscript presented in an intelligible fashion and written in standard English?

Reviewer #1: Yes

Reviewer #2: Yes

6. Review Comments to the Author

Reviewer #1: I have no additional comment/question. Although the model is interesting, I have some doubts regarding the ability of this model to predict response to therapies in PPGL, since there is no HIF2 alpha and no expected elevated succinate..

Reviewer #2: The authors addressed the comments satisfactorily, in accordance with the reviewers comments and the manuscript is now in good shape.

7. PLOS authors have the option to publish the peer review history of their article (what does this mean?). If published, this will include your full peer review and any attached files.

Reviewer #1: No

Reviewer #2: No

---

## [Editor Report · Acceptance letter]

17 Dec 2019

PONE-D-19-29380R1 

Modeling succinate dehydrogenase loss disorders in *C. elegans* through effects on hypoxia-inducible factor 

Dear Dr. Maher, III:

I am pleased to inform you that your manuscript has been deemed suitable for publication in PLOS ONE. Congratulations! Your manuscript is now with our production department. 

With kind regards,

on behalf of

Dr. Ales Vicha 

Academic Editor

PLOS ONE